# Research on the evolution and driving factors of the coupling relationship among tourism industry, urbanization and human settlements of the urban agglomeration in the middle reaches of the Yangtze River

**Youbao Yang[1], Ailiang Xie[2]\*, Xin Zhou[3]\*, Kang Cheng[4]**

**1** School of Tourism, Hunan Normal University, Changsha, China, **2** School of History and Culture, Linyi University, Linyi, China, **3** School of Public Administration, Central South University, Changsha, China, **4** School of Business, Hunan Normal University, Changsha, China

\* xieailiang@lyu.edu.cn (AX); zstar0926@163.com (XZ)

**Data Availability Statement:** All relevant data are within the paper and its Supporting Information files.

## Abstract

This study examines the evolution of the coupling relationship among the tourism industry, urbanization, and human settlements of the urban agglomeration in the middle reaches of the Yangtze River from 2005 to 2019. By employing a constructed evaluation index system and a quantitative analysis model, the study aims to characterize and analyze this relationship. The key findings are as follows: (1) The coupling coordination level of the three major systems exhibits a continuous upward trend over the study period, albeit with a gradually slowing growth rate. The Changsha -Zhuzhou-Xiangtan urban agglomeration shows the highest average growth rate, followed by the Wuhan metropolitan area and the Poyang Lake urban agglomeration. (2) The variation coefficient and Taylor index of the coupling coordination degree demonstrate a fluctuating downward trend, indicating a convergence in spatial differences in interregional coupling coordination levels. Internal differences between city clusters within urban agglomerations are identified as the main source of overall differences in coupling coordination levels. (3) The coupling coordination levels of the three major systems follow an evolutionary trajectory characterized by "serious imbalance, moderate imbalance, mild imbalance". Clear features of level transitions are observed among various cities. (4) The endowment of tourism resources emerges as the primary factor driving the evolution of the coupling relationship among the tourism industry, urbanization, and human settlements. The study highlights the increasing diversification of driving forces and significant spatiotemporal heterogeneity in interregional driving forces for coupling relationship evolution factors.

## Introduction

In the contemporary landscape of socioeconomic development, the intricate relationship among the tourism industry, urbanization, and the quality of human settlements has emerged

**Funding:** This study was supported by the Philosophy and Social Science Foundation of China "Research on the dynamic mechanism and realization paths of high-quality development of cultural tourism integration in relatively poor areas of China" (Grant no. 21BGL150).

**Competing interests:** All authors of the paper, including Dr. Youbao Yang, Prof Ailiang Xie, Dr. Xin Zhou and Dr. Kang Cheng declare no conflict of interest.

as a critical determinant of regional prosperity. The tourism industry characterized by its dynamism and global reach, stands as a cornerstone of economic activity, driving not only revenue generation but also social and cultural transformations [1]. Simultaneously, urbanization, propelled by technological advancements and demographic shifts, plays a pivotal role in shaping the spatial and social fabric of regions. It enhances infrastructure, services, and innovation ecosystems, thus amplifying the allure of urban areas for tourists and residents alike [2]. Moreover, the quality of human settlements, encompassing aspects of infrastructure, environmental sustainability, and community well-being, serves as a linchpin for sustainable development and tourism competitiveness [3]. Additionally, whether the interaction between the tourism industry and urbanization has negative effects needs to be further explored. While existing research highlights the positive correlation and mutual benefits, potential adverse impacts such as environmental degradation, social displacement, and infrastructure strain have not been thoroughly examined. This study aims to address this gap by investigating both the positive and negative consequences of this interplay, thereby providing a more balanced and comprehensive understanding of the dynamics at play.

However, amidst the promise of progress, China faces a host of challenges in aligning the trajectories of its tourism industry, urbanization, and human settlements. Discrepancies and inadequacies in regional tourism development, urbanization patterns, and living conditions pose obstacles to the nation's vision of high-quality economic and social development [1]. This discordance underscores the need for a nuanced understanding of the interplay between these domains and strategic interventions to harmonize their evolution.

Against this backdrop, this study aims to delve into the intricate dynamics of the tourism industry, urbanization, and human settlements in China's context, with a particular focus on urban agglomeration in the middle reaches of the Yangtze River. By elucidating the evolution patterns and driving factors shaping the coupling relationship among these components from 2005 to 2019, the research seeks to uncover opportunities for enhancing regional development strategies. The paper is structured as follows: it begins with an overview of the research objectives, proceeds to delineate the methodology employed, presents the findings, and concludes with a discussion of implications and avenues for future research. Through this comprehensive analysis, we endeavor to contribute to the discourse on regional development and inform policy formulation aimed at fostering sustainable growth in China and beyond. Presently, academic research on the intricate interplay among the tourism industry, urbanization, and human settlements has made substantial strides, yielding noteworthy findings. Scholarship in this realm has predominantly concentrated on the subsequent facets: In 1991, Mullins introduced the notion of 'tourism urbanization', a seminal milestone that instigated the examination of the nexus between tourism and urbanization [4]. Subsequently, international scholars have delved into the theoretical dimensions, encompassing the attributes, typologies, mechanisms, and models of tourism urbanization [5–8]. In parallel, the rapid expansion of China's tourism sector and the swifter pace of urbanization have drawn the attention of domestic academic circles, prompting scrutiny into the phenomenon and theoretical considerations related to tourism urbanization [9–11]. The empirical research findings concerning the interplay between tourism and urbanization have been steadily emerging. A widely ratified consensus within the academic community affirms the existence of a positive interaction between tourism development and urbanization, substantiating a robust correlation between the two [12–14]. The second focal area of investigation revolves around the relationship among the tourism industry and human settlements. On the global front, researchers have adopted a holistic approach, integrating both qualitative and quantitative research methodologies, to elucidate the interrelation between these elements. This involves examinations from the perspective of human-land dynamics [15], the confluence of tourism and urban development [16], and the

nexus between tourism and regional sustainability [17]. On the domestic front, the emphasis has shifted towards quantitative assessments of the quality of human settlements in tourist destinations through the selection of representative case studies and an in-depth analysis of pertinent influencing factors. This research has uncovered the pivotal role of tourism development in guiding the transformation of human settlements in tourist locales [18]. Moreover, through the construction of quantitative analytical models, it has dissected the coupling and coordination dynamics between the tourism industry and human settlements [19,20], scrutinizing the relationship among tourism allure and the quality of human settlements. The third dimension of inquiry pertains to the relationship among urbanization and human settlements [21]. Building upon the 'human settlement' theory introduced by Sadias, Western academia has engaged in extensive investigations into facets such as the comprehensive evaluation of human settlements [22] and the interplay between human settlements and the quality of life and well-being [23]. In parallel, buoyed by Wu Liangyong's 'human settlements science', domestic scholarship has extended its focus towards the direct interrelationship between urbanization and human settlements, culminating in the conclusion that there exists a dual-sided dynamic of mutual promotion and antagonism between urbanization and human settlements, underscoring the stress-laden dichotomy [24,25]. Against the backdrop of China's burgeoning urbanization and the mounting challenges within human settlements contexts, the overarching themes of 'enhancing urbanization quality' and 'improving human settlements' have evolved into pivotal components of the national socioeconomic development strategy in the contemporary era [26].

Within the corpus of existing academic literature, both domestic and international, significant advancements have been made in exploring the intricate relationship among the tourism industry, urbanization, and human settlements. This prior work serves as a valuable foundation for the continued investigation in this paper, however, certain shortcomings persist and warrant careful consideration and enhancement. Primarily, extant research predominantly addresses the interplay between the tourism industry, urbanization, and human settlements. Nevertheless, from a systemic perspective, there is a notable dearth of comprehensive investigations into the symbiotic coupling dynamics among these three domains. Existing research has yet to encompass the tourism industry, urbanization, and human settlements within an integrated and unified theoretical framework. Consequently, the coupling mechanisms among these elements and the determinants of their interrelationships remain insufficiently elucidated. Secondly, prevailing research efforts have primarily concentrated on localized spatial scales, often limited to individual cities or provinces. Consequently, examinations of the spatial and temporal heterogeneity within the coupling and coordination relationships among intercity tourism industry-urbanization-human settlements, particularly at the scale of urban agglomerations, are relatively underrepresented. This discrepancy is misaligned with the contemporary strategic imperatives for the coordinated spatial development of urban agglomerations in China. To address these gaps in the literature, this study adopts urban agglomeration in the middle reaches of the Yangtze River as a case study area. It treats the triad of the tourism industry, urbanization, and human settlements as an intricately interrelated, structurally comprehensive, and cyclically interconnected macro-system. Drawing upon theoretical explorations of the coupling mechanisms among these components, the study introduces quantitative models such as coupling coordination degrees, coefficient of variation, Taylor index, and Geodetector model to delineate and analyze the spatial and temporal evolution characteristics and driving factors underlying the coupling relationships among the three dimensions. The overarching aim is to furnish a theoretical underpinning and decision-making insights for the systematic advancement of the synergistic connections between the tourism industry,

urbanization, and human settlements within urban agglomerations, thereby facilitating the realization of high-quality economic and social development within these urban clusters.

The harmonious coexistence of the human-land relationship has long been a central focus of academic inquiry. Empirical evidence and theoretical frameworks underscore the existence of a natural coupling relationship among the tourism industry, urbanization, and human settlements. These three components collectively form a complex and open circulatory system. A pivotal task is the profound comprehension and analysis of the coupling mechanisms underpinning these three components, representing the fundamental challenge in elucidating the developmental dynamics between regional tourism industry, urbanization, and human settlements. To elucidate this coupling mechanism, we deconstruct it as delineated in Fig 1.

First and foremost, the tourism industry assumes a critical role as a potent driving force and a novel catalyst for advancing urbanization and ameliorating human settlements. Its salient attributes, including a robust correlation-driven effect, high input-output efficiency, and minimal ecological environmental pollution, render it instrumental in these transformations. On one hand, through the development of tourism resources, the stimulation of tourism consumption demand, and the generation of economic benefits, the tourism industry bolsters the economic aggregate of tourist destinations. This leads to the acceleration of industrial restructuring, the creation of employment opportunities, and the catalyzing of rural population urbanization. Consequently, it engenders the expansion of urban spatial realms, the qualitative enhancement of urban landscapes, the restructuring of land use, and the promotion of economic, social, population, and spatial urbanization within tourist destinations. On the

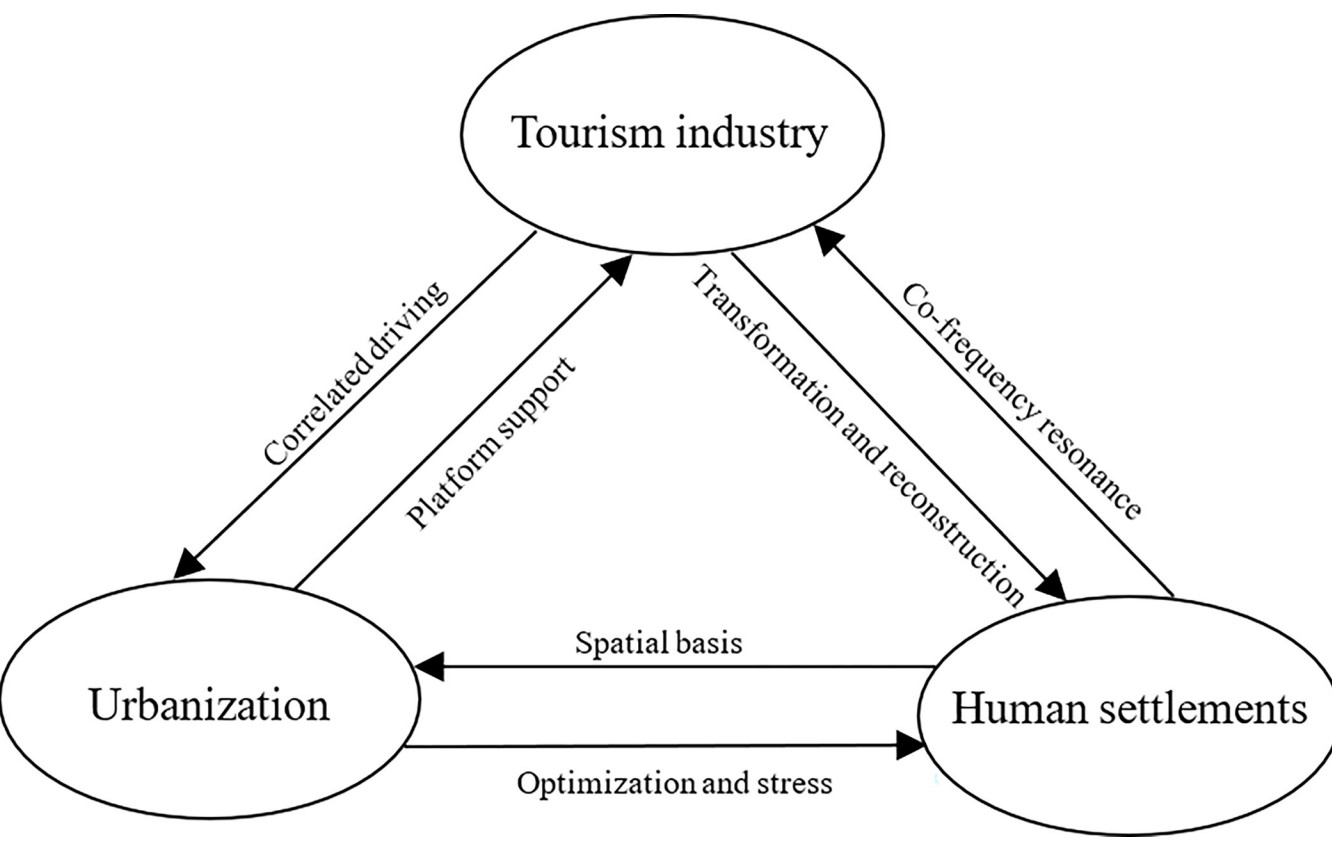

**Fig 1. The coupling mechanism among tourism industry, urbanization and human settlements.**

other hand, regional tourism development has a transformative effect on human settlements. The development of tourism resources, the expansion of the tourism market, and the augmentation of economic benefits provide a robust economic and material foundation for the evolution of human settlements. The surge in demand for ancillary service facilities during the course of tourism development propels simultaneous enhancements in residents' daily living conditions, public infrastructure, ecological landscape environments, education, culture, and health undertakings in tourist destinations. As a result, tourism development intervention emerges as a formidable force influencing the evolution and reconstruction of human settlements, becoming a pivotal mode for the transformation and development of urban and rural spaces in the contemporary era.

Secondly, urbanization, as a dynamic historical process emblematic of advanced human social evolution, serves as a vital platform that drives tourism industry development and guides the transformation of human settlements. Urbanization contributes to the continuous growth of the tourism economy, playing a pivotal role in advancing the transformation and upgrade of the tourism sector. By steering economic and social advancement, enhancing population quality, and expanding regional space, urbanization furnishes a framework for enhancing tourism infrastructure, boosting investment in tourism development, stimulating tourism consumption, and expanding tourism capacity. Urbanization, especially within the urbanization process, plays a defining role in shaping the tourism industry's ability to meet factors such as "food, housing, transportation, travel, purchase, and entertainment," thereby substantially influencing the growth of the tourism sector. Concurrently, urbanization, as a harbinger of evolving lifestyles, entails a parallel process of constructing human settlements. The evolution of human settlements represents a tangible manifestation of the urbanization effect, serving as an indicator of the dynamic spatiotemporal changes in economics, society, culture, ecology, and spatial configurations resulting from urbanization. Urbanization exerts a dual effect on human settlements, with a propensity for positive optimization and, at times, creating stresses on the latter. Hence, the quality of urban human settlements serves as a crucial gauge for assessing the scientific and orderly nature of urbanization. The enhancement of human settlements quality has emerged as a key objective of urbanization.

Thirdly, human settlements represent the foundational spaces for human existence and habitation. A favorable human settlements is a coveted urban resource and a cornerstone for augmenting the appeal and competitiveness of urban tourism while continuing the urbanization process. On one hand, urban human settlements and tourism industry activities exhibit a pronounced "resonance". Elevated human settlements provide a pivotal foundation for reshaping the city's external image, enhancing tourism market allure, and augmenting tourism receptiveness. In an era marked by upgraded social consumption demands, tourists no longer fixate solely on traditional natural and cultural tourism assets; dimensions of human settlements, including urban economy, society, culture, and ecology, have become integral considerations in tourists' decision-making processes. Human settlements quality now stands as a fundamental criterion for measuring the developmental potential of urban tourism. On the other hand, a salubrious human settlements underpins the trajectory of urbanization. As human social civilization progresses, the quest for a harmonious and habitable urban environment has become a fundamental aspiration for modern life. The optimization and renewal of the human settlements constitute a cardinal mission in contemporary urban development. The augmentation of human settlements quality bolsters modern cities' contributions to human well-being and social progress, providing a solid spatial foundation for stimulating urban economic and social development, attracting populations to cities, and enhancing social and cultural atmospheres. In tandem, it forges a potent backdrop for fostering economic urbanization, social urbanization, population urbanization, and spatial urbanization.

While statistical comparisons were made among urbanization, human settlements, and the tourism industry in the study, it is crucial to acknowledge the potential independence of factors contributing to urbanization and human settlements dynamics in these cities. The analysis accounted for various influencing factors beyond the scope of the tourism industry, recognizing the multifaceted nature of urban development. This underscores the importance of considering the broader context and potential interplay of factors influencing urbanization and human settlements patterns. It is recommended that future research adopt a more comprehensive approach to examine all potential drivers of urban development, rather than solely focusing on the tourism industry. This could involve extensive data collection and nuanced analysis to gain a holistic understanding of the drivers behind urbanization and human settlements evolution.

## Materials and methods

### Entropy method

In this study, the entropy method is used to evaluate the comprehensive quality level of tourism industry, urbanization and human settlements of the urban agglomeration in the middle reaches of the Yangtze River. Through standardization, normalization, calculation of information entropy, redundancy and other steps, the weight value of each index is obtained, and then the comprehensive evaluation index of the three is calculated [27,28]. The formula is:

$$S_{\lambda i} = \sum_{j=1}^{n} W_j \times X_{\lambda ij}$$

In the formula: $S_{\lambda i}$ is the comprehensive evaluation index of tourism industry, urbanization and human settlements of the $i$ city in the $\lambda$ year, $W_j$ is the weight value of the $j$ index, $X_{\lambda ij}$ is the standardized value of the $j$ index of the $i$ city in the $\lambda$ year, and $n$ is the number of evaluation index systems of tourism industry, urbanization and human settlements.

### Coupling coordination degree model

Based on the coupling theory in physics [29], the measurement model of the coupling and coordination relationship among tourism industry, urbanization and human settlements is introduced. The calculation formula is as follows:

$$C = \left\{ \frac{U_1 \times U_2 \times U_3}{[(U_1 + U_2 + U_3)/3]^3} \right\}^{1/3}$$

$$D = \sqrt{C \times T}$$

$$T = \alpha U_1 + \beta U_2 + \gamma U_3$$

In the formula: $C$ is the coupling degree, $U_1$, $U_2$, $U_3$ respectively, represent the comprehensive evaluation index of tourism industry, urbanization and human settlements, $0 \leq C \leq 1$, when $C = 1$, means that the three are in the best coupling state, when $C = 0$, the three are in an unrelated state and develop in a disorderly direction; $D$ is the coupling coordination degree represents the overall coordination effect of the three, and $\alpha$, $\beta$, $\gamma$ is the weight coefficient of the three. Considering that the development of the tourism industry is an important driving force for urbanization and the optimization of the human settlements, its dynamic evolution will have a correlation impact on the other two systems. So, set $\alpha = 0.4$, $\beta = 0.3$, $\gamma = 0.3$. At the same time, the coupling coordination degree is divided into 10 levels by using the "0.1 segmentation cut-off point method "and

summarized into three types: imbalance, transition and coordination. The specific division criteria are referred to the relevant literature [30], which is not repeated here.

## Coefficient of variation and Taylor index

The coefficient of variation [31] and Taylor index [32] are used to measure the regional differences in the coupling and coordination level of tourism industry-urbanization-human settlements. These two indexes are often used to measure the imbalance or difference of factor development. At present, they have been widely used in the spatial difference analysis of geography. The calculation formula is:

$$CV = 1\overline{h} \sqrt{\frac{1}{n} \sum_{i=1}^{n} \left( h_i - \overline{h} \right)^2}$$

In the formula: $CV$ is the coefficient of variation; $n$ is the number of sample cities for the study ($n$ = 28 taken in this article, the same below) ; $h_i$ is the coupling coordination degree of the sample city $i$ (the same below) ; $\overline{h}$ is the average value of $h_i$ (the same below). The larger the coefficient of variation, the greater the degree of fluctuation of the coupling coordination level, and the greater the regional difference.

The Taylor index can decompose the overall difference of coupling coordination level into the differences within and between the three zones of Wuhan metropolitan area, Poyang Lake urban agglomeration and Changsha-Zhuzhou-Xiangtan urban agglomeration.

$$Theil = \frac{1}{n} \sum_{i=1}^{n} \frac{h_i}{\overline{h}} ln \frac{h_i}{\overline{h}} = T_{WR} + T_{BR}$$

$$T_{WR} = \sum_{k=1}^{k} \frac{n_k}{n} * \frac{h_k}{\overline{h}} \left( \frac{1}{n} \sum_{i \in gk} \frac{h_i}{h_k} ln \frac{h_i}{h_k} \right) = \sum_{k=1}^{k} \frac{n_k}{n} * \frac{h_k}{\overline{h}} * T(h_k)$$

$$T_{BR} = \sum_{k=1}^{k} \frac{n_k}{n} * \frac{h_k}{\overline{h}} * ln \frac{h_k}{\overline{h}}$$

In the formula: $Theil$ is the overall difference; $T_{WR}$ is differences within the three zones; $T_{BR}$ is the difference between the three major zones; $k$ is the number of zone groupings in the region ($k$ = 3 taken in this paper), $n_k$ is the number of sample cities grouped in the $k$ zone and $h_k$ is the average number of coupling coordination degrees grouped in the $k$ zone.

## Geodetector model

Geodetector is a new statistical method to detect spatial heterogeneity and explain the driving factors behind it [33]. The Geodetector includes four parts: risk detection, factor detection, ecological detection and interactive detection. This study mainly uses factor detection and interactive detection to analyze the driving factors of the evolution of the coupling relationship among tourism industry, urbanization and human settlements of the urban agglomeration in the middle reaches of the Yangtze River. The factor detection formula is as follows:

$$q = 1 - \frac{1}{N\sigma^2} \sum_{m=1}^{L} N_m \sigma_m^2$$

In the formula: $q$ is the driving force of a certain factor on the evolution of the coupling relationship among the tourism industry-urbanization-human settlements, $L$ is the number of classification layers, $N_m$, $N$ represent the number of units of the layer m and the research area, $\sigma_m{}^2$ and $\sigma^2$ represents the discrete variance of the layer m and the entire research area. The $q$ value range is [0,1], and the larger the value is, the stronger the driving force of this factor on the evolution of the coupling relationship among tourism industry-urbanization-human settlements is.

## Indicator system and data sources

In order to quantitatively evaluate the comprehensive development level of the three major systems of tourism industry, urbanization and human settlements, on the basis of following the principles of index representativeness, data availability and system relevance, and referring to relevant research results [12,19], from the 11 dimensions of tourism resource endowment, tourism market demand, tourism economic benefit, economic urbanization, social urbanization, population urbanization, space urbanization, living and living conditions, ecological landscape environment, public infrastructure, science, education, culture and health, 43 index systems were selected to quantitatively measure the comprehensive development level of the three, as shown in Table 1.

The basic data of this study are mainly derived from the 《*Hunan Statistical Yearbook*》 (2006–2020), 《*Hubei Statistical Yearbook, Jiangxi Statistical Yearbook*》 (2006–2020), 《*China Urban Statistical Yearbook*》 (2006–2020) 《*China Urban Construction Statistical Yearbook*》 (2005–2019) and the statistical bulletins of the national economic and social development of each city (2005–2019). Before analysis, the collected data underwent preprocessing steps to ensure consistency and accuracy. Missing data points in individual years were supplemented using interpolation methods based on adjacent years' data. This approach helps maintain the temporal integrity of the dataset while minimizing the impact of missing values on the analysis.

## Research area

The urban agglomeration in the middle reaches of the Yangtze River spans three provinces such as Hubei, Hunan and Jiangxi, in order to facilitate the analysis and refer to the existing research, the research area further categorized into three urban agglomerations: the Wuhan metropolitan area, the Changsha-Zhuzhou-Xiangtan urban agglomeration, and the Poyang Lake urban agglomeration. This classification allows for a more nuanced analysis of the coupling coordination degree and its evolution over time, highlighting the spatial heterogeneity within the broader region.

## The overall evolution trend of coupling coordination degree of the urban agglomeration in the middle reaches of the Yangtze River

Employing the entropy method and the coupling coordination degree model, the longitudinal evolution trend of the coupling coordination degree within the tourism industry-urbanization-human settlements triad of the urban agglomeration in the middle reaches of the Yangtze River from 2005 to 2019 has been meticulously analyzed and is depicted in Fig 2. This comprehensive examination of the data reveals that since 2005, the coupling coordination level among the regional tourism industry, urbanization, and human settlements has demonstrated a consistent upward trajectory. The coupling coordination degree has surged from 0.1807 to 0.3333, reflecting an impressive growth rate of approximately 84.49%. This pattern underscores that as the scale of the tourism industry has expanded, the level of urbanization has progressed, and

**Table 1. Comprehensive evaluation index system of tourism industry, urbanization and human settlements.**

| System layer | Dimensional layer | Indicator layer | The nature of indicators | Weight |
|---|---|---|---|---|
| Tourism industry system (T) | Tourism resource endowment ($T_1$) | $T_{11}$/The high-grade tourist attraction[①] (piece) | + | 0.0292 |
| | | $T_{12}$/National forest park (piece) | + | 0.0413 |
| | | $T_{13}$/National wetland park (piece) | + | 0.0700 |
| | | $T_{14}$/National key cultural relics protection units (piece) | + | 0.0372 |
| | | $T_{15}$/Traditional Chinese villages (piece) | + | 0.0118 |
| | Tourist demand ($T_2$) | $T_{21}$/Number of domestic tourists (Unit/10000) | + | 0.0162 |
| | | $T_{22}$/Inbound tourists (Unit/10000) | + | 0.0145 |
| | | $T_{23}$/Distribution density of tourist arrivals (10000/km$^2$) | + | 0.0170 |
| | Tourist economic benefit ($T_3$) | $T_{31}$/Domestic tourism revenue (Billion yuan) | + | 0.0168 |
| | | $T_{32}$/Inbound tourism revenue (million USD) | + | 0.0121 |
| | | $T_{33}$/Total tourism revenue as a proportion of GDP (%) | + | 0.0239 |
| Urbanization system (U) | Economic urbanization ($U_1$) | $U_{11}$/Per capital GDP (yuan) | + | 0.0328 |
| | | $U_{12}$/Proportion of secondary and tertiary industries in GDP (%) | + | 0.0292 |
| | | $U_{13}$/Local fiscal self-sufficiency rate[②] (%) | + | 0.0163 |
| | | $U_{14}$/The actual amount of foreign capital used in that year (million USD) | + | 0.0153 |
| | Social urbanization ($U_2$) | $U_{21}$/Total retail sales of consumer goods (million yuan) | + | 0.0174 |
| | | $U_{22}$/Number of registered urban unemployed (person) | - | 0.0130 |
| | | $U_{23}$/Average wage of urban on-the-job workers (yuan) | + | 0.0404 |
| | | $U_{24}$/Balance of year-end savings deposits of urban and rural residents (million yuan) | + | 0.0213 |
| | Population urbanization ($U_3$) | $U_{31}$/Urban population distribution density (person/km$^2$) | + | 0.0455 |
| | | $U_{32}$/The proportion of urban population in the total population (%) | + | 0.0453 |
| | | $U_{33}$/The natural growth rate of urban population (‰) | + | 0.0096 |
| | | $U_{34}$/Proportion of employees in secondary and tertiary industries (%) | + | 0.0073 |
| | Space urbanization ($U_4$) | $U_{41}$/Built-up area (km$^2$) | + | 0.0203 |
| | | $U_{42}$/Built-up area expansion rate (%) | + | 0.0045 |
| | | $U_{43}$/Construction land area (km$^2$) | + | 0.0183 |
| | | $U_{44}$/The proportion of construction land area in urban area (%) | + | 0.0273 |
| Human settlements system (E) | Living conditions ($E_1$) | $E_{11}$/Residential land area per capita (m$^2$/person) | + | 0.0165 |
| | | $E_{12}$/Completed residential development investment (million yuan) | + | 0.0171 |
| | | $E_{13}$/Water penetration rate (%) | + | 0.0063 |
| | | $E_{14}$/Gas penetration rate (%) | + | 0.0120 |
| | Ecological landscape environment ($E_2$) | $E_{21}$/Green coverage rate of built district (%) | + | 0.0103 |
| | | $E_{22}$/Per capita public green space area (m$^2$) | + | 0.0198 |
| | | $E_{23}$/Centralized Treatment Rate of Sewage Treatment Plant (%) | + | 0.0368 |
| | | $E_{24}$/Harmless treatment ratio for house refuse (%) | + | 0.0235 |
| | Public infrastructure ($E_3$) | $E_{31}$/Per capita urban road area (m$^2$/person) | + | 0.0155 |
| | | $E_{32}$/At the end of the year, the number of bus (electric) vehicles in operation was actual (Vehicles) | + | 0.0328 |
| | | $E_{33}$/Length of pipes (km) | + | 0.0158 |
| | | $E_{34}$/Electricity consumption of urban and rural residents (10000kw/h) | + | 0.0195 |
| | Science, education, culture and health undertakings ($E_4$) | $E_{41}$/The proportion of science and technology expenditure in local fiscal expenditure (%) | + | 0.0084 |
| | | $E_{42}$/Number of students in ordinary colleges and universities per ten thousand people (person) | + | 0.0592 |
| | | $E_{43}$/Public library book collections (Thousand volumes) | + | 0.0262 |
| | | $E_{44}$/Number of hospital beds (Sheet) | + | 0.0260 |

Note: ①High-level tourist attractions, this article mainly refers to 4A and 5A tourist attractions; ②Local fiscal self-sufficiency rate = local fiscal general budget revenue / local fiscal general budget expenditure.

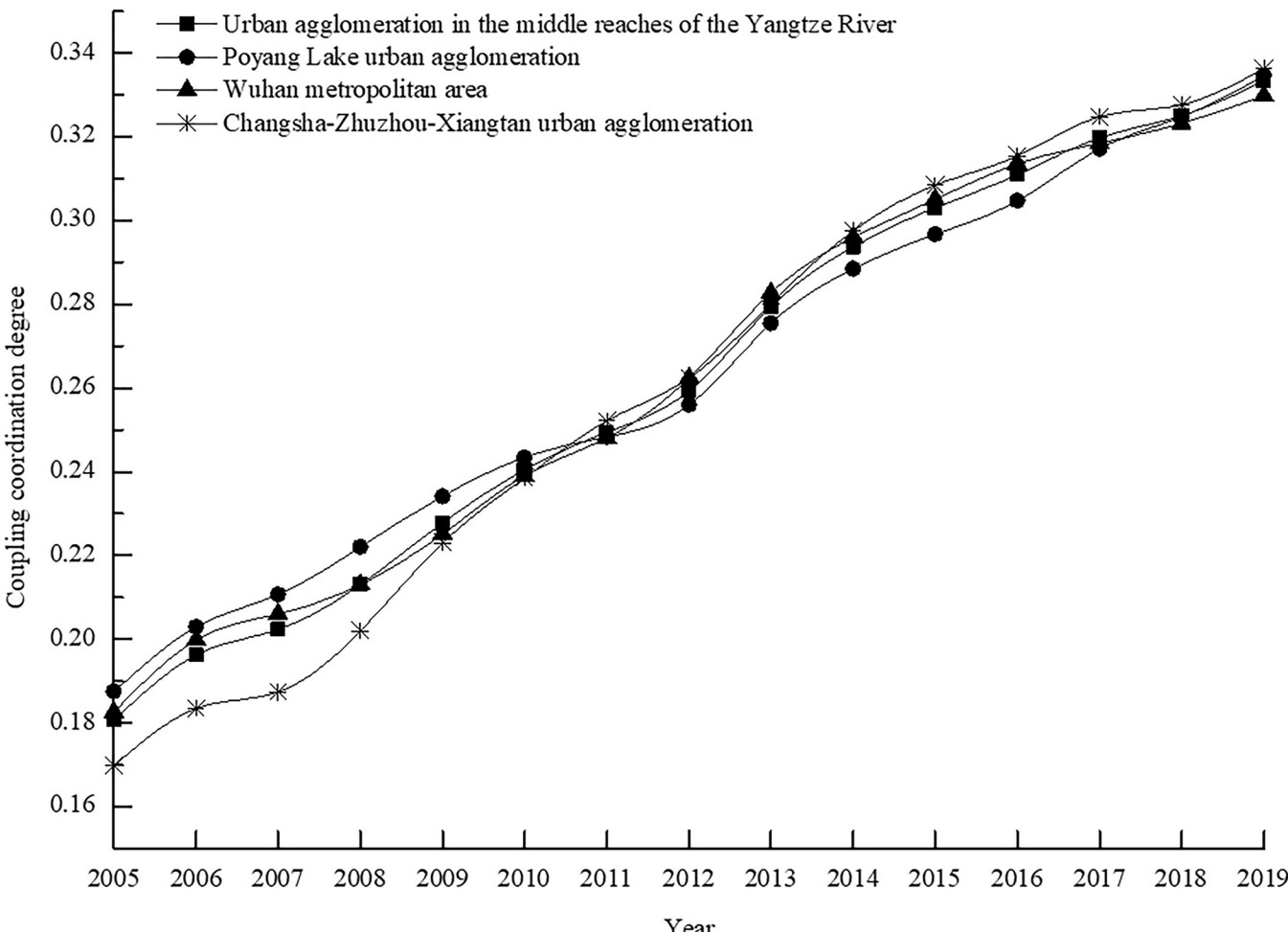

**Fig 2. The overall evolution trend of the coupling coordination degree of tourism industry-urbanization-human settlements from 2005 to 2019.**

the quality of human settlements within urban agglomerations has persistently improved, the synergistic correlation effects among these three components have intensified progressively.

Nevertheless, it is noteworthy that during the study period, the growth rate of the coupling coordination degree among the three elements has gradually decelerated. The year 2011 emerges as a pivotal inflection point in this trend. The average annual growth rate of the coupling coordination degree witnessed a decline from 5.53% during the period 2006–2011 to 3.71% during the years 2012–2019. Upon a thorough examination of the original data, it becomes evident that this deceleration can be attributed primarily to the inception of the 12th Five-Year Plan. This plan marked a significant juncture wherein China embarked on a transformative path to revamp its economic growth model, promote the construction of an "ecological civilization," and elevate the national strategy for new urbanization. The period witnessed a heightened focus on the intrinsic qualities and structural optimization of tourism industry growth, the pursuit of healthy and orderly urbanization, and the enhancement of the quality of urban human settlements construction. These strategic pivots slowed the pace of comprehensive development in urban agglomerations concerning the tourism industry, urbanization, and human settlements. Consequently, the growth rate of the coupling coordination degree among these three components exhibited a synchronous downturn.

Examining the evolution trend of the coupling coordination degree across the three principal zone units within the urban agglomeration, namely the Wuhan metropolitan area, the Changsha-Zhuzhou-Xiangtan urban agglomeration, and the Poyang Lake urban agglomeration, reveals a consistent upward trajectory with gradually decelerating growth rates, mirroring the overarching trend of the urban agglomeration in the middle reaches of the Yangtze River. Notably, the Changsha-Zhuzhou-Xiangtan urban agglomeration displayed the most rapid growth rate, with an average annual increase of 5.04% from 2005 to 2019. Since 2011, this urban agglomeration has gradually surged to the forefront among the three units. This surge signifies that, under the strategic impetus of robust tourism city development, metropolitan area integration, and the establishment of "two-oriented society" demonstration zones, the evolution of the tourism industry, urbanization, and human settlements transformation within the Changsha-Zhuzhou-Xiangtan urban agglomeration has yielded noteworthy outcomes. Conversely, the Wuhan metropolitan area and the Poyang Lake urban agglomeration exhibit nearly identical growth rates, posting respective average annual growth rates of 4.34% and 4.24% from 2005 to 2019. These two zones demonstrate relatively limited growth potential in the coupling coordination degree of the tourism industry-urbanization-human settlements. Their journey towards optimizing the coupling coordination relationship among these three elements is still fraught with challenges, and the untapped potential for enhancing the coupling coordination level warrants further exploration and realization.

Analyzing the evolution trend of the coefficient of variation for the coupling coordination degree (as illustrated in Fig 3), we observe a fluctuating downward trajectory of the urban agglomeration in the middle reaches of the Yangtze River and the three principal areas, collectively spanning the years 2005 to 2019. This trend signifies that the spatial divergence in the coupling coordination levels among the tourism industry, urbanization, and human settlements within the urban agglomeration and the three major areas is diminishing. Moreover, the degree of variability in the coupling coordination degree is attenuating, indicative of an increasing level of harmony and synchronization among these three components.

Among these regions, the coefficient of variation exhibits the most pronounced fluctuations within the Changsha-Zhuzhou-Xiangtan urban agglomeration, witnessing a decrease of approximately 42.67% over the period spanning 2005 to 2019. This underscores the volatility in the evolution of the coupling and coordination relationship among the tourism industry, urbanization, and human settlements. The spatial disparities in the coupling and coordination levels across this region vary considerably, with the coefficient of variation for the coupling coordination degree in the Poyang Lake urban agglomeration consistently residing at the lowest level, signifying an enduring trend of stability. This phenomenon reflects the shrinking spatial disparities in the coupling coordination levels among the tourism industry, urbanization, and human settlements within this zone, indicating that the evolution of the coupling coordination levels between cities is progressively converging.

Conversely, the coefficient of variation for the coupling coordination degree within the Wuhan metropolitan area frequently holds the top rank among the three primary zones in most years, displaying a rebound trend within the overarching downward trajectory. This dynamic attests to the marked spatial disparities in the coupling coordination levels among the tourism industry, urbanization, and human settlements within this zone. Strong spatial heterogeneity persists in the inter-city coupling coordination relationship.

Based on the computation results of the Taylor index for the coupling coordination degree (illustrated in Fig 4), it is discernible that the Taylor index exhibits a consistent "staircase-like" decline over the span of 2005 to 2019. The index value registers a decrement from 0.0186 to 0.0093, translating to a notable reduction of about 49.66%. This pattern underscores that the spatial disparities in the coupling coordination degree among the tourism industry,

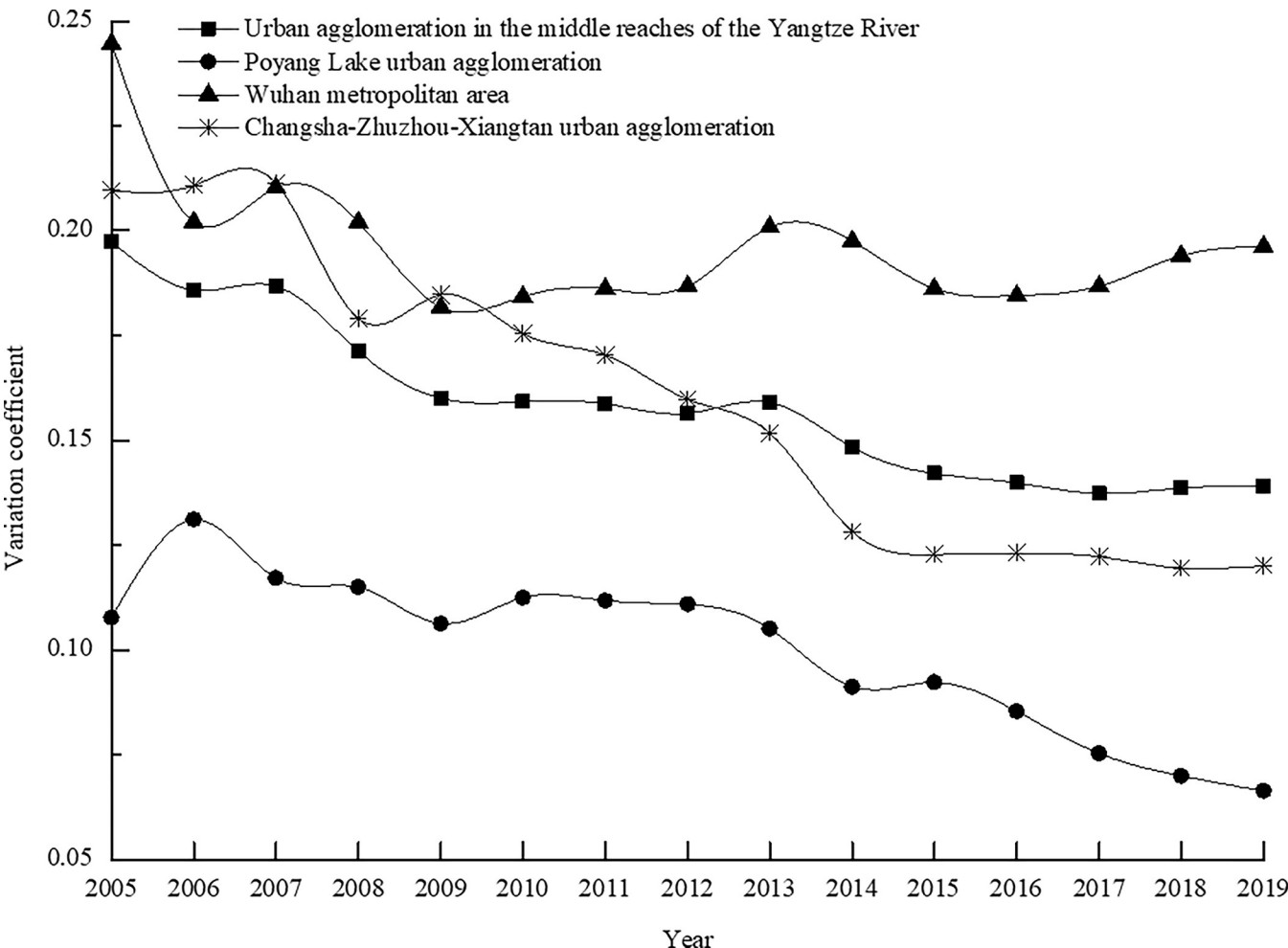

**Fig 3. The evolution trend of the coefficient of variation of the coupling coordination degree of tourism industry-urbanization-human settlements from 2005 to 2019.**

urbanization, and human settlements of the urban agglomeration in the middle reaches of the Yangtze River have gradually contracted during the study period. Moreover, the spatial convergence in the coupling coordination level among these three components has continually strengthened, corroborating earlier research findings. The evolution trend in intra-regional disparities closely aligns with the overarching disparities, signifying a substantial degree of overlap between the two. Hence, it becomes evident that the divergence in the coupling coordination levels of the tourism industry, urbanization, and human settlements within the three major zones of the urban agglomeration is synchronously diminishing.

By comparing disparities in the coupling and coordination levels both between and within regions, it becomes apparent that from 2005 to 2019, intra-regional disparities have been more pronounced than inter-regional disparities. The average contribution rates of the former and the latter to the overall disparities are 98.06% and 1.94%, respectively. This disparity in contribution rates highlights that the primary source of the overall disparities in the coupling and coordination degree among the tourism industry, urbanization, and human settlements of the urban agglomeration in the middle reaches of the Yangtze River stems from intra-regional disparities. Specifically, the variations of the coupling and coordination levels within the three

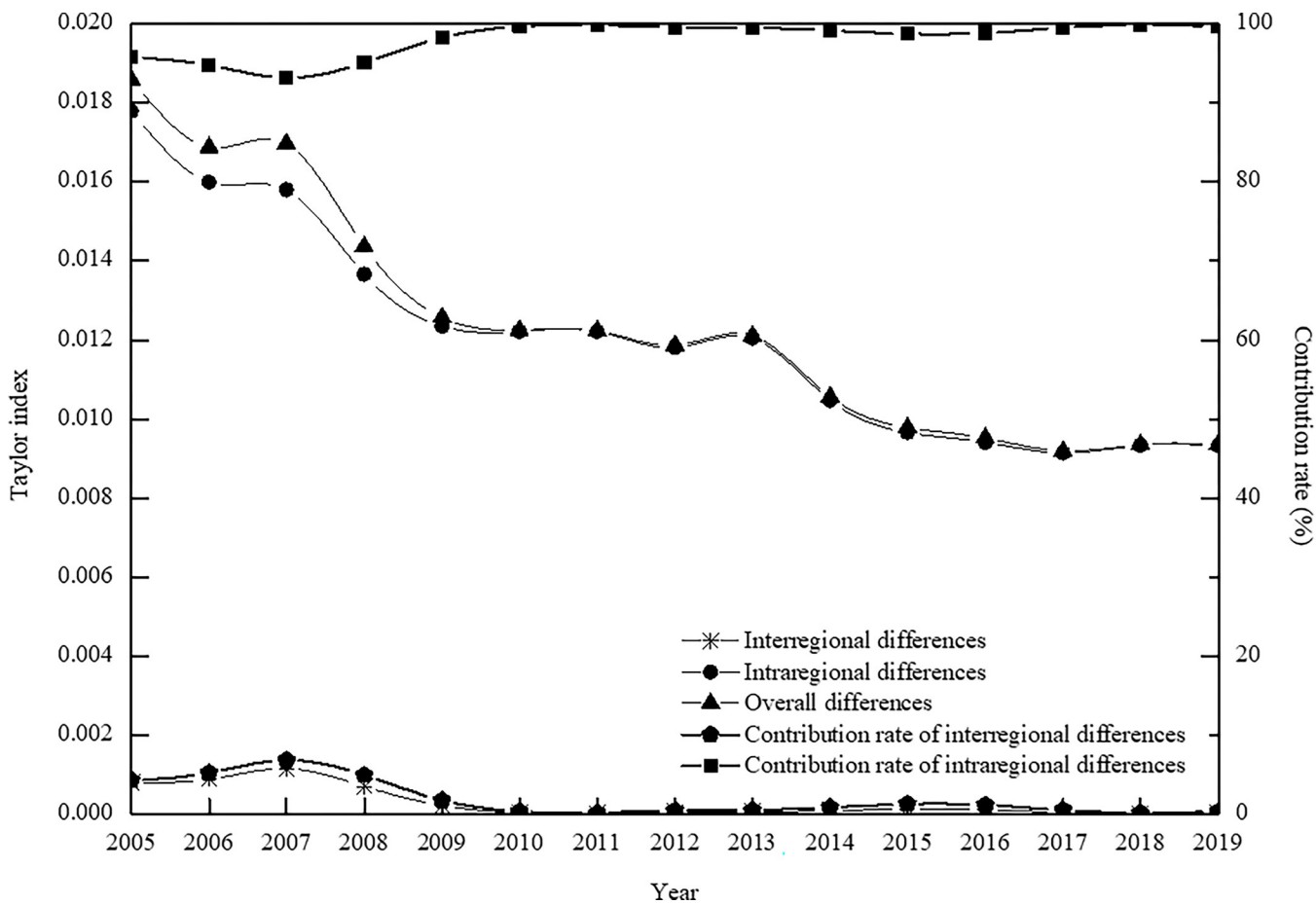

**Fig 4. The evolution trend of Taylor index of coupling coordination degree of tourism industry-urbanization-human settlements from 2005 to 2019.**

regions have the following characteristics: the Wuhan metropolitan area, the Changsha-Zhuzhou-Xiangtan urban agglomeration, and the Poyang Lake urban agglomeration constitute the principal wellspring of disparities in the overall urban agglomeration. Addressing these intraregional disparities represents a pivotal objective for optimizing the overarching coupling and coordination relationships of the urban agglomeration in the middle reaches of the Yangtze River. The endeavor is to continually narrow the developmental gap in the coupling and coordination levels among the tourism industry, urbanization, and human settlements within the three major regions.

## Spatio-temporal evolution characteristics of coupling coordination level of the urban agglomeration in the middle reaches of the Yangtze River

Based on the calculated results of the coupling coordination degree and in adherence to the pertinent division standards [30], we have obtained a comprehensive perspective on the evolution of the coupling coordination levels pertaining to the tourism industry, urbanization, and human settlements within the urban agglomeration and the three principal zones located in the middle reaches of the Yangtze River from 2005 to 2019. As displayed in Table 2, since 2005, the evolution path of the coupling coordination level within these regions has

**Table 2. The coupling coordination level distribution of tourism industry-urbanization-human settlements from 2005 to 2019.**

| Area | 2005 | 2006 | 2007 | 2008 | 2009 | 2010 | 2011 | 2012 | 2013 | 2014 | 2015 | 2016 | 2017 | 2018 | 2019 |
|---|---|---|---|---|---|---|---|---|---|---|---|---|---|---|---|
| Urban agglomeration in the middle reaches of the Yangtze river | I | II | II | II | II | II | II | II | II | III | III | III | III | III | III |
| Poyang Lake urban agglomeration | I | II | II | II | II | II | II | II | II | II | III | III | III | III | III |
| Wuhan metropolitan area | I | II | II | II | II | II | II | II | II | III | III | III | III | III | III |
| Changsha-Zhuzhou-Xiangtan urban agglomeration | I | I | I | II | II | II | II | II | II | III | III | III | III | III | III |

Note: I represent severe imbalance; II represents moderate disorder; III represents mild disorders.

consistently traversed a trajectory characterized by a transition from "serious imbalance" to "moderate imbalance," ultimately converging towards "mild imbalance." This pattern signifies that the coupling coordination level of the regional tourism industry, urbanization, and human settlements has demonstrated a continual enhancement. The interplay and coordination dynamics among these three components have witnessed marked improvement. However, it is imperative to acknowledge that no pivotal shift in the coupling coordination type has occurred during the study period. The regions remain situated within the realm of "imbalance" development. Consequently, the coupling coordination level of the tourism industry, urbanization, and human settlements within the urban agglomeration persists at a relatively modest level.

Within this context, it is notable that the temporal evolution patterns characterizing the coupling coordination levels of the Wuhan metropolitan area and the urban agglomeration in the middle reaches of the Yangtze River exhibit a high degree of synchronization. Conversely, the temporal evolution of the coupling coordination levels within the urban agglomerations encompassing Changsha-Zhuzhou-Xiangtan and Poyang Lake lags behind comparatively. The latter two major urban agglomerations made the transition from "serious imbalance" to "moderate imbalance" and subsequently to "mild imbalance" in 2008 and 2015, respectively. These transitions occurred markedly later than the corresponding events in the Wuhan metropolitan area and the urban agglomeration in the middle reaches of the Yangtze River.

Furthermore, with regard to the distribution frequency of the coupling coordination levels pertaining to the tourism industry, urbanization, and human settlements, the most prevalent classification is the "moderate imbalance" level, boasting a substantial frequency of 51.67%. This finding underscores that within the overarching strategic framework of advancing the high-quality development of the tourism industry, nurturing the paradigm of new urbanization, and advancing the construction of an ecological civilization, the urban agglomeration in the middle reaches of the Yangtze River is still confronted with the pressing imperative of elevating the quality and advancing the enhancement of the coupling coordination level of the tourism industry, urbanization, and human settlements. The interplay and coordination dynamics among these three components have yet to reach the desired operational state. To further elucidate the spatial evolution characteristics of the coupling and coordination level within the tourism industry, urbanization, and human settlements of the urban agglomeration situated in the middle reaches of the Yangtze River, an analysis was conducted from the perspective of city scale, with 2005, 2012, and 2019 serving as key time points.

As portrayed in **Table 3,** it becomes apparent that since 2005, the coupling coordination level of the tourism industry-urbanization-human settlements of the urban agglomeration situated in the middle reaches of the Yangtze River exhibits conspicuous grade transition characteristics. The region is characterized by four coupling coordination levels such as "on the verge of imbalance", "mild imbalance", "moderate imbalance", and "severe imbalance". This intricate relationship evolves from a state of "imbalance" to a "transition state". Nevertheless, the overall coupling coordination level at the regional level remains relatively low, aligning with previous

**Table 3. Evolution pattern of coupling coordination level of tourism industry-urbanization-human settlements from 2005 to 2019.**

| Year | Coupling Coordination Level | Subordinate City |
|------|------------------------------|------------------|
| 2005 | Serious Dissonance | Changde, Jingzhou, Jingmen, Xiaogan, Huanggang, Ezhou, Huangshi, Xianning, Yueyang, Yiyang, Loudi, Hengyang, Xiangtan, Zhuzhou, Pingxiang, Ji'an, Xinyu, Yichun, Nanchang, Fuzhou, Yingtan |
| | Moderate Dissonance | Xiangyang, Yichang, Wuhan, Changsha, Jiujiang, Jingdezhen, Shangrao |
| | Mild Dissonance | None |
| | Borderline Dissonance | None |
| 2012 | Serious Dissonance | Loudi |
| | Moderate Dissonance | Xiangyang, Jingmen, Xiaogan, Huanggang, Ezhou, Huangshi, Xianning, Jingzhou, Changde, Yueyang, Yiyang, Xiangtan, Zhuzhou, Hengyang, Pingxiang, Xinyu, Yichun, Ji'an, Nanchang, Fuzhou, Jingdezhen, Yingtan |
| | Mild Dissonance | Yichang, Wuhan, Jiujiang, Changsha, Shangrao |
| | Borderline Dissonance | None |
| 2019 | Serious Dissonance | None |
| | Moderate Dissonance | Ezhou, Huangshi, Loudi |
| | Mild Dissonance | Xiangyang, Yichang, Jingmen, Xiaogan, Huanggang, Xianning, Jingzhou, Changde, Yueyang, Yiyang, Xiangtan, Zhuzhou, Hengyang, Pingxiang, Xinyu, Yichun, Ji'an, Nanchang, Fuzhou, Jiujiang, Jingdezhen, Shangrao, Yingtan |
| | Borderline Dissonance | Wuhan, Changsha |

research findings. More specifically: In 2005, only two levels, namely "moderate imbalance" and "severe imbalance," were observed in the coupling and coordination relationship among the tourism industry, urbanization, and human settlements within the region. The cities within these two levels exhibited a ratio of 1:3. Notably, the "moderate imbalance" level was solely represented by cities such as Jingdezhen, Jiujiang, Shangrao, Wuhan, Yichang, Xiangyang, and Changsha. This observation underscores the prevailing notion that the synergistic effect among the tourism industry, urbanization, and human settlements of the urban agglomeration situated in the middle reaches of the Yangtze River was generally not robust. Additionally, there existed a degree of decoupling development among these three components. In 2012, a substantial shift was observed in the regional coupling coordination level. Loudi City remained the sole representative of the "serious imbalance" category, while most other cities transitioned to either the "moderate imbalance" or "mild imbalance" levels. Approximately 89.29% of cities elevated their coupling levels from low to high. The shift in grade predominantly manifested as "mild imbalance" in cities like Jiujiang, Shangrao, Wuhan, Yichang, and Changsha. In 2019, yet another change occurred in the regional coupling and coordination level. The "serious imbalance" category was entirely phased out, and the "moderate imbalance" level was only preserved in cities like Huangshi, Ezhou, and Loudi. The majority of cities upgraded to either "mild imbalance" or the "on the verge of imbalance" level. Approximately 82.14% of cities transitioned from low coupling levels to high coupling levels. The "on the verge of imbalance" level was only present in Wuhan and Changsha. The spatial distribution of the coupling and coordination level of the tourism industry, urbanization, and human settlements within the urban agglomeration conforms to a typical "center-periphery" structural pattern.

## Driving factors for the evolution of the coupling relationship among tourism industry, urbanization and human settlements

Utilizing the factor detection and interactive detection module within the Geodetector model, we further examine the driving factors behind the evolution of the coupling relationship

among the tourism industry, urbanization, and human settlements of the urban agglomeration situated in the middle reaches of the Yangtze River over the period spanning from 2005 to 2019. The development of the coupling and coordination relationship among the three components has demonstrated a relatively stable trend since 2005. This analysis accounts for multiple index factors and years, focusing on an eleven-dimensional framework. Through the manual classification and discretization of independent variables in 2005, 2012, and 2019, the Geodetector software platform was employed to identify eleven pivotal driving factors for factor detection and interactive detection. The following summarizes the results of the Geodetector:

The factor detection module yielded results regarding the driving forces influencing the evolution of the coupling relationship among the tourism industry, urbanization, and human settlements of the urban agglomeration situated in the middle reaches of the Yangtze River. To better comprehend the changing trends of the detected factors in the time series, these factors are ranked according to their q-values (Table 4).

In 2005, the paramount factor shaping the coupling relationship among the tourism industry, urbanization, and human settlements of the urban agglomeration situated in the middle reaches of the Yangtze River was the endowment of tourism resources, with a notably high q-value of 0.3417. The factor ranked first with a factor degree of 1.97, signifying the foundational role that tourism resource endowment plays in bolstering the development of the regional tourism industry. It emerged as a pivotal driver guiding the evolution of the coupling relationship among the tourism industry, urbanization, and human settlements, while the influence of other factors remained relatively modest.

In 2012, tourism resource endowment continued to be the primary driving factor, with a q-value surge to 0.4327. Concurrently, the impacts of tourism market demand, science, education, culture, health, space urbanization, and other factors were amplified, with these factors moving up in the rankings.

By 2019, tourism resource endowment still retained its primary factor status, with the q-value escalating to 0.4932. The q-values of economic urbanization, public infrastructure, social urbanization, and other factors experienced substantial growth, and their rankings advanced accordingly. This evolution in the coupling relationship indicates that over time, factors such as urban comprehensive economic strength, public infrastructure development level, and urban socialization have gained increasing influence. The diversity in driving forces shaping

**Table 4. Detection results of evolution factors of coupling relationship among tourism industry, urbanization and human settlements from 2005 to 2019.**

| 2005 | | 2012 | | 2019 | |
|---|---|---|---|---|---|
| Factor order | q | Factor order | q | Factor order | q |
| Tourism resource endowment ($T_1$) | 0.3417 | Tourism resource endowment ($T_1$) | 0.4327 | Tourism resource endowment ($T_1$) | 0.4932 |
| Science, education, culture and health undertakings ($E_4$) | 0.1733 | Tourist demand ($T_2$) | 0.2332 | Economic urbanization ($U_1$) | 0.3350 |
| Economic urbanization ($U_1$) | 0.1682 | Culture and health undertakings ($E_4$) | 0.2216 | Public infrastructure ($E_3$) | 0.3068 |
| Population urbanization ($U_3$) | 0.1662 | Space urbanization ($U_4$) | 0.1493 | Social urbanization ($U_2$) | 0.2994 |
| Tourist demand ($T_2$) | 0.1619 | Economic urbanization ($U_1$) | 0.0799 | Culture and health undertakings ($E_4$) | 0.2863 |
| Ecological landscape environment ($E_2$) | 0.084 | Social urbanization ($U_2$) | 0.0695 | Tourist economic benefit ($T_3$) | 0.2554 |
| Public infrastructure ($E_3$) | 0.0825 | Tourist economic benefit ($T_3$) | 0.0602 | Space urbanization ($U_4$) | 0.1589 |
| Living conditions ($E_1$) | 0.0386 | Ecological landscape environment ($E_2$) | 0.0295 | Living conditions ($E_1$) | 0.0364 |
| Space urbanization ($U_4$) | 0.0373 | Living conditions ($E_1$) | 0.0264 | Tourist demand ($T_2$) | 0.0196 |
| Tourist economic benefit ($T_3$) | 0.0229 | Public infrastructure ($E_3$) | 0.0213 | Population urbanization ($U_3$) | 0.0053 |
| Social urbanization ($U_2$) | 0.0191 | Population urbanization ($U_3$) | 0.0111 | Ecological landscape environment ($E_2$) | 0.0002 |

the coupling relationship among the tourism industry, urbanization, and human settlements became more pronounced.

When examining the time series, it's notable that the q-values of five key driving factors, namely tourism resource endowment, tourism economic benefits, social urbanization, spatial urbanization, and science, education, culture, and health undertakings, consistently exhibited a steady upward trajectory. This trend reflects the pivotal conditions related to regional tourism resources, the comprehensive development of the tourism economy, the process of urban socialization, the scale of regional space, and the significant role of public utilities encompassing science, education, culture, and health in driving the development of the tourism industry, the promotion of urbanization, and the establishment of the human settlements. As a result, these variables emerged as pivotal contributors impacting the evolution of the coupling relationship among the regional tourism industry, urbanization, and human settlements. From a sub-regional perspective, there is notable spatial and temporal heterogeneity in the driving factors behind the evolution of the coupling relationship among the tourism industry, urbanization, and human settlements in the Wuhan metropolitan area, Changsha-Zhuzhou-Xiangtan urban agglomeration, and the Poyang Lake urban agglomeration since 2005. This heterogeneity is summarized in Table 5.

In the Wuhan metropolitan area, the primary driving factor from 2005 to 2019 follows a path of "tourism resource endowment—spatial urbanization—economic urbanization" succession. During this study period, the q-value and primacy degree of the aforementioned factor exhibited a declining trend. This observation suggests that the dominance of the primary factor in driving the evolution of the coupling relationship among the tourism industry, urbanization, and human settlements in the Wuhan metropolitan area has gradually weakened since 2005, signifying a trend toward diversified driving forces. The most substantial reduction in factor influence q-value was observed for tourism resource endowment, with a decline of up to 59.08%. In contrast, the influence of economic urbanization, tourism economic benefits, ecological landscape environment, and living conditions showed a significant upward trend.

For the Changsha-Zhuzhou-Xiangtan urban agglomeration, the primary driving factors indicate a path of succession involving "ecological landscape environment—tourism resource endowment." The primacy of these factors exhibits an evolutionary trend, initially rising and then declining. Notably, tourism resource endowment, spatial urbanization, tourism economic benefits, living conditions, and ecological landscape environment are the most influential factors. The overall ranking of factor q-values or rankings is on the rise.

In the Poyang Lake urban agglomeration, the primary driving factor has consistently been the endowment of tourism resources. The first degree and q-value of this factor are increasing, and the gap with other factors is widening. The driving factors shaping the evolution of the coupling relationship among the regional tourism industry, urbanization, and human settlements tend to be monopolistic and singular. In this context, factors such as tourism economic benefits, science, education, culture, and health, as well as living conditions, have seen a notable decrease in q-values, accompanied by a consistent shift in rankings towards lower positions. This trend indicates a weakening influence of these factors on the evolution of the coupling relationship among the regional tourism industry, urbanization, and human settlements.

## Conclusions and discussions

### Conclusions

Upon examining the coupling mechanism of the tourism industry, urbanization, and human settlements, this study established an evaluation index system and introduced quantitative

**Table 5. Detection results of evolution factors of coupling relationship among tourism industry, urbanization and human settlements in different regions from 2005 to 2019.**

| Detection factor | 2005 | | | 2012 | | | 2019 | | |
|---|---|---|---|---|---|---|---|---|---|
| | Wuhan metropolitan area | Changsha-Zhuzhou-Xiangtan urban agglomeration | Poyang Lake urban agglomeration | Wuhan metropolitan area | Changsha-Zhuzhou-Xiangtan urban agglomeration | Poyang Lake urban agglomeration | Wuhan metropolitan area | Changsha-Zhuzhou-Xiangtan urban agglomeration | Poyang Lake urban agglomeration |
| Tourism resource endowment ($T_1$) | 0.8186 | 0.0476 | 0.4038 | 0.5473 | 0.7504 | 0.6667 | 0.3350 | 0.6000 | 0.5050 |
| Tourist demand ($T_2$) | 0.0212 | 0.0476 | 0.0962 | 0.4444 | 0.0274 | 0.0018 | 0.3103 | 0.2381 | 0.0594 |
| Tourist economic benefit ($T_3$) | 0.0476 | 0.0476 | 0.4038 | 0.1770 | 0.0274 | 0.0385 | 0.4655 | 0.2889 | 0.2203 |
| Economic urbanization ($U_1$) | 0.4286 | 0.2000 | 0.0462 | 0.4074 | 0.0014 | 0.1827 | 0.4828 | 0.0857 | 0.1419 |
| Social urbanization ($U_2$) | 0.0212 | 0.0476 | 0.2564 | 0.5650 | 0.3678 | 0.0385 | 0.3103 | 0.0044 | 0.1213 |
| Population urbanization ($U_3$) | 0.1012 | 0.1333 | 0.0183 | 0.0329 | 0.0695 | 0.1026 | 0.0345 | 0.2000 | 0.1749 |
| Space urbanization ($U_4$) | 0.0212 | 0.4286 | 0.0962 | 0.5679 | 0.0638 | 0.0385 | 0.4828 | 0.5556 | 0.0570 |
| Living conditions ($E_1$) | 0.0476 | 0.0222 | 0.0962 | 0.1512 | 0.0071 | 0.4631 | 0.3103 | 0.2889 | 0.0099 |
| Ecological landscape environment ($E_2$) | 0.2744 | 0.4444 | 0.2564 | 0.0672 | 0.0638 | 0.0865 | 0.4636 | 0.2889 | 0.1859 |
| Public infrastructure ($E_3$) | 0.3386 | 0.0476 | 0.0962 | 0.3964 | 0.0071 | 0.2428 | 0.3103 | 0.0857 | 0.1859 |
| Culture and health undertakings ($E_4$) | 0.3386 | 0.4286 | 0.4038 | 0.3964 | 0.3678 | 0.0505 | 0.3103 | 0.0222 | 0.1213 |

analysis models such as the entropy method, coupling coordination degree, coefficient of variation, Taylor index, and Geodetector. The research aimed to describe and analyze the evolution characteristics and driving factors behind the coupling relationship among these three elements of the urban agglomeration situated in the middle reaches of the Yangtze River from 2005 to 2019. The study findings are as follows:

The coupling and coordination level of the tourism industry, urbanization, and human settlements of the urban agglomeration situated in the middle reaches of the Yangtze River exhibited a continuous upward trend, signifying an increasing synergistic effect among the three. However, the growth rate of the coupling coordination degree is gradually slowing down. The year 2011 marks a pivotal time point in this growth rate transition. The evolutionary trend of the coupling and coordination relationship across the three urban agglomerations is in line with the overall trend, with the average growth rate ranking as follows: Changsha-Zhuzhou-

Xiangtan urban agglomeration > Wuhan metropolitan area > Poyang Lake urban agglomeration.

The coefficient of variation and the Taylor index of the coupling coordination degree for the tourism industry, urbanization, and human settlements of the urban agglomeration situated in the middle reaches of the Yangtze River, as well as in the three major areas, display a fluctuating downward trend. This suggests that the regional coupling coordination degree's dispersion is weakening, and the spatial difference in the coupling coordination level among the three elements is contracting. In this context, the difference within the urban agglomeration regions has consistently been greater than the differences among the regions, making the intra-regional differences the main contributor to the overall variance in the coupling coordination degree for the tourism industry, urbanization, and human settlements.

The coupling coordination level of the tourism industry, urbanization, and human settlements of the urban agglomeration situated in the middle reaches of the Yangtze River and the three major areas follows an evolution trajectory characterized by "serious imbalance, moderate imbalance, and mild imbalance." The regional coupling coordination level continues to rise, and the interaction and coordination among the three elements are improving. Notably, there is a pronounced shift in the coupling coordination level within various cities, involving four grades: on the verge of imbalance, mild imbalance, moderate imbalance, and severe imbalance. The transition from an "imbalance state" to a "transition state" is a defining characteristic of this coupling coordination type.

The primary driving factor influencing the evolution of the coupling relationship among the tourism industry, urbanization, and human settlements of the urban agglomeration situated in the middle reaches of the Yangtze River is tourism resource endowment. Over time, the influence of factors such as tourism economic benefits, social urbanization, spatial urbanization, science, education, culture, and health services has been steadily increasing. This diversification in driving forces indicates that the primary driving factors in the evolution of the coupling relationship are becoming more diverse. Specific to each region, the Wuhan metropolitan area follows a path of "tourism resource endowment—space urbanization—economic urbanization" as its primary driving factors. In contrast, the primary driving factor for the Changsha-Zhuzhou-Xiangtan urban agglomeration shifts from ecological landscape environment to tourism resource endowment. The Poyang Lake urban agglomeration maintains tourism resource endowment as its predominant factor.

## Discussion and implications

The findings underscore the importance of understanding the intricate interdependencies among the tourism industry, urbanization, and human settlements for formulating comprehensive and effective regional development strategies. Recognizing these interdependencies allows policymakers to grasp the holistic picture of regional dynamics, enabling them to design targeted interventions that address the specific needs and challenges of each component while considering their synergistic effects. Policymakers should not only recognize the evolving dynamics of coupling coordination but also proactively anticipate future trends and challenges.

By prioritizing interventions that foster balanced and sustainable development across urban agglomerations, policymakers can mitigate potential disparities and maximize the positive impacts of development initiatives. This approach involves promoting inclusive growth strategies that ensure equitable access to opportunities, resources, and services for all segments of the population.

Policymakers should adopt a proactive stance towards enhancing resilience and adaptability in the face of external shocks and uncertainties, such as economic fluctuations, environmental

changes, and technological disruptions. This may involve investing in infrastructure, fostering innovation ecosystems, and strengthening social safety nets to build robust and resilient urban systems capable of withstanding and recovering from various challenges.

Policymakers should engage in stakeholder collaboration and participatory decision-making processes to ensure the alignment of development goals and priorities with the needs and aspirations of local communities. By fostering transparent and inclusive governance structures, policymakers can enhance trust, legitimacy, and accountability, ultimately leading to more effective implementation and sustainable outcomes.

## Research prospects

While this research contributes significant insights into the complex interplay among the tourism industry, urbanization, and human settlements within urban agglomerations, it is essential to acknowledge that these three systems operate both independently and in concert, forming a web of interconnected dynamics that shape regional development trajectories. Despite the progress made in understanding their synergistic relationship, the theoretical exploration of their coupling remains in its infancy, leaving ample room for further investigation and refinement.

Future research endeavors should delve deeper into several key areas. Firstly, there is a need to expand and refine the evaluation index system and quantitative analysis models to capture the multifaceted nature of the coupling relationship comprehensively. This entails incorporating additional indicators that reflect the diverse dimensions of tourism development, urbanization processes, and human settlements dynamics, thereby providing a more nuanced understanding of their interdependencies.

Efforts should be made to precisely identify the coupling and coordination relationships among the three major systems and their dynamic evolution over time. By employing advanced analytical techniques and methodologies, researchers can discern the underlying mechanisms driving the interactions between tourism, urbanization, and human settlements, unraveling the intricate patterns of their co-evolution and identifying critical tipping points and feedback loops.

Future research should adopt an interdisciplinary perspective, drawing insights from fields such as economics, geography, environment, and management to develop holistic optimization strategies. By integrating knowledge and methodologies from diverse disciplines, researchers can devise innovative approaches to address the complex challenges facing urban agglomerations, fostering sustainable and inclusive development pathways that maximize societal welfare and environmental resilience.

In the study, we did not provide detailed explanations on how to ensure other conditions remain constant when comparing the tourism industry, urbanization, and human settlements factors of cities. Specifically, to associate urbanization and human settlements factors with the tourism industry, the tourism potential of each city must be equivalent. We acknowledge that this could potentially impact the interpretation of the study results. We intend to address this by considering the use of standardized assessment criteria or controlling for other influencing variables in future research to ensure the reliability and validity of the results. This point needs to be articulated and addressed more clearly in our study.

While this study sheds light on the interplay of the tourism industry, urbanization, and human settlements within urban agglomerations, it represents only a preliminary step towards unraveling the complexities of their coupling dynamics. By addressing the aforementioned research gaps and adopting a multidisciplinary approach, future studies can contribute to a more nuanced understanding of the intricate relationships shaping regional development

trajectories, ultimately informing more effective policy interventions and decision-making processes.

## Supporting information

**S1 File.**
(XLSX)

## Author Contributions

**Conceptualization:** Youbao Yang, Xin Zhou.

**Data curation:** Youbao Yang.

**Formal analysis:** Youbao Yang, Xin Zhou.

**Investigation:** Xin Zhou.

**Methodology:** Kang Cheng.

**Software:** Xin Zhou.

**Validation:** Xin Zhou.

**Writing – original draft:** Ailiang Xie, Kang Cheng.

**Writing – review & editing:** Ailiang Xie, Xin Zhou.

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
