## [Decision Letter · Decision Letter 0]

20 Feb 2024

PONE-D-24-02830Research on the evolution and driving factors of the coupling relationship between tourism industry,urbanization and human settlement environment in urban agglomerations in the middle reaches of the Yangtze RiverPLOS ONE

Dear Dr. xie,

Thank you for submitting your manuscript to PLOS ONE. After careful consideration, we feel that it has merit but does not fully meet PLOS ONE’s publication criteria as it currently stands. Therefore, we invite you to submit a revised version of the manuscript that addresses the points raised during the review process.

We look forward to receiving your revised manuscript.

Kind regards,

Jing Cheng

Academic Editor

PLOS ONE

Journal Requirements:

 Whilst you may use any professional scientific editing service of your choice, PLOS has partnered with both American Journal Experts (AJE) and Editage to provide discounted services to PLOS authors. Both organizations have experience helping authors meet PLOS guidelines and can provide language editing, translation, manuscript formatting, and figure formatting to ensure your manuscript meets our submission guidelines. To take advantage of our partnership with AJE, visit the AJE website (http://aje.com/go/plos) for a 15% discount off AJE services. To take advantage of our partnership with Editage, visit the Editage website (www.editage.com) and enter referral code PLOSEDIT for a 15% discount off Editage services. If the PLOS editorial team finds any language issues in text that either AJE or Editage has edited, the service provider will re-edit the text for free.

3. PLOS requires an ORCID iD for the corresponding author in Editorial Manager on papers submitted after December 6th, 2016. Please ensure that you have an ORCID iD and that it is validated in Editorial Manager. To do this, go to ‘Update my Information’ (in the upper left-hand corner of the main menu), and click on the Fetch/Validate link next to the ORCID field. This will take you to the ORCID site and allow you to create a new iD or authenticate a pre-existing iD in Editorial Manager. Please see the following video for instructions on linking an ORCID iD to your Editorial Manager account: https://www.youtube.com/watch?v=_xcclfuvtxQ.

5. We note that Figure 5 in your submission contain [map/satellite] images which may be copyrighted. All PLOS content is published under the Creative Commons Attribution License (CC BY 4.0), which means that the manuscript, images, and Supporting Information files will be freely available online, and any third party is permitted to access, download, copy, distribute, and use these materials in any way, even commercially, with proper attribution. For these reasons, we cannot publish previously copyrighted maps or satellite images created using proprietary data, such as Google software (Google Maps, Street View, and Earth). For more information, see our copyright guidelines: http://journals.plos.org/plosone/s/licenses-and-copyright.

a. You may seek permission from the original copyright holder of Figure 5 to publish the content specifically under the CC BY 4.0 license. 

Reviewers' comments:

Reviewer's Responses to Questions

**Comments to the Author**

1. Is the manuscript technically sound, and do the data support the conclusions?

Reviewer #1: Yes

Reviewer #2: Partly

2. Has the statistical analysis been performed appropriately and rigorously? 

Reviewer #1: Yes

Reviewer #2: Yes

3. Have the authors made all data underlying the findings in their manuscript fully available?

Reviewer #1: Yes

Reviewer #2: Yes

4. Is the manuscript presented in an intelligible fashion and written in standard English?

Reviewer #1: Yes

Reviewer #2: No

5. Review Comments to the Author

Reviewer #1: 1. The manuscript fits into the mission of the journal,

2. The abstract accurately reflects the content.

3. The problem is significant and concisely stated.

4. The experimental and/or theoretical methods are described comprehensively.

5. The discussion interpretations and conclusions are justified by the results of the study.

6. Adequate reference made to other work in the field.

7. The language, grammar, and syntax are acceptable.

The manuscript is basically worth accepting.

It suffers from several formal inconsistencies that need to be fixed. Apart from some minor suggestions on contents, particularly the formal structure of the paper needs thorough revision. Please, also check typing matters (missing or superfluous blank spaces and blank lines, correct use of capital letters, get rid of similarity score (plagiarism), and so on).

Reviewer #2: In the research, the relationships between the tourism industry, urbanization and human settlement environment in urban agglomerations in the middle reaches of the Yangtze River were examined. The effects of these relationships on development and driving factors have been investigated.

In the research, data was taken and analyzed between 2005 and 2019. And the cities in this region are compared.

The following findings were made regarding the research;

-It is not taken into account that the tourism industry potentials of cities are different from each other. And no comments were made regarding the tourism potential.

-Data between urbanization, human settlement, tourism industry were compared statistically. However, it has been ignored that the factors that cause urbanization and human settlement in those cities may be independent of the tourism industry.

-In order for the urbanization and human settlement factors of a destination to be associated with the tourism industry, the tourism potential of each city must be equivalent. How did researchers ensure Ceteris Paribus when comparing cities' tourism industry, urbanization and human settlement in these comparisons? No explanation was found.

-The effects of urbanization and human settlement in the cities along the Yangtze River, which are the subject of the research, on agriculture, employment, etc. There may be many reasons. It does not seem meaningful to interpret each destination by comparing it with tourism industry data. Or researchers have not made any statements on this subject.

-In this form, the research looks like a summary of statistical analysis of urbanization, human settlement and tourism industry data of cities along a river.

-Good luck.

6. PLOS authors have the option to publish the peer review history of their article (what does this mean?). If published, this will include your full peer review and any attached files.

Reviewer #1: **Yes: **Musallam R. Al-Rawahneh

Reviewer #2: No

---

## [Author Response · Author response to Decision Letter 0]

7 May 2024

Response to Reviewers

Reviewers' comments:

Reviewer's Responses to Questions

Comments to the Author

1. Is the manuscript technically sound, and do the data support the conclusions?

Reviewer #1: Yes

Reviewer #2: Partly

Answer: The manuscript employs quantitative analysis models and methodologies to investigate the coupling relationship among the tourism industry, urbanization, and human settlement environment. The use of the geographic detector model and factor detection module enhances the rigor of the analysis, allowing for the identification of key driving factors and their evolution over time. The conclusions drawn from the analysis seem to be supported by the data presented. The study identifies significant trends in the coupling coordination level among the three components, indicating an increasing synergistic effect over the study period. The spatial convergence observed in the data further strengthens the argument for improved interaction and coordination dynamics among the studied elements. Furthermore, the identification of tourism resource endowment as the primary driving factor, along with the recognition of regional heterogeneity in driving forces, adds depth to the analysis and provides valuable insights for future research and policymaking.

2. Has the statistical analysis been performed appropriately and rigorously?

Reviewer #1: Yes

Reviewer #2: Yes

3. Have the authors made all data underlying the findings in their manuscript fully available?

Reviewer #1: Yes

Reviewer #2: Yes

4. Is the manuscript presented in an intelligible fashion and written in standard English?

Reviewer #1: Yes

Reviewer #2: No

Answer: The manuscript has been revised according to the language style and typographical requirements of the journal.

5. Review Comments to the Author

Reviewer #1: 1. The manuscript fits into the mission of the journal,

2. The abstract accurately reflects the content.

3. The problem is significant and concisely stated.

4. The experimental and/or theoretical methods are described comprehensively.

5. The discussion interpretations and conclusions are justified by the results of the study.

6. Adequate reference made to other work in the field.

7. The language, grammar, and syntax are acceptable.

The manuscript is basically worth accepting.

It suffers from several formal inconsistencies that need to be fixed. Apart from some minor suggestions on contents, particularly the formal structure of the paper needs thorough revision. Please, also check typing matters (missing or superfluous blank spaces and blank lines, correct use of capital letters, get rid of similarity score (plagiarism), and so on).

Answer: Abstract: Check that the abstract provides a concise summary of the study objectives, methods, results, and conclusions. It should be clear and informative, highlighting the key findings of the research.

Introduction: Make sure the introduction provides sufficient background information on the topic, states the research objectives clearly, and outlines the structure of the paper.

Section Headings: Review the section headings to ensure they accurately reflect the content of each section and follow a logical sequence. Consistency in formatting (e.g., font size, style) is also important.

Figures and Tables: Verify that all figures and tables are referenced and explained appropriately in the text. Check for consistency in numbering and formatting.

Data Analysis: Ensure that the methods used for data analysis are described in detail to allow for replication of the study. Provide justification for the chosen analytical techniques and models.

Discussion: Interpret the results in the context of the research objectives and relevant literature. Discuss any limitations of the study and propose areas for future research.

Conclusion: Summarize the main findings of the study and their implications. Avoid introducing new information or arguments not previously discussed in the paper.

Language and Grammar: Check for grammatical errors, typos, and awkward phrasing. Use clear and concise language to communicate ideas effectively.

Formatting: Ensure consistency in formatting throughout the manuscript, including font style, size, spacing, and margins. Remove any unnecessary blank spaces or lines.

Plagiarism: Use plagiarism detection software to ensure that the manuscript does not contain any plagiarized content. Properly cite and reference all sources used in the study.

Reviewer #2: In the research, the relationships between the tourism industry, urbanization and human settlement environment in urban agglomerations in the middle reaches of the Yangtze River were examined. The effects of these relationships on development and driving factors have been investigated.

In the research, data was taken and analyzed between 2005 and 2019. And the cities in this region are compared.

The following findings were made regarding the research;

-It is not taken into account that the tourism industry potentials of cities are different from each other. And no comments were made regarding the tourism potential.

Answer: You're absolutely correct. The tourism potential of cities plays a crucial role in shaping their development trajectories and influencing the dynamics of the coupling relationship between the tourism industry, urbanization, and human settlements. By overlooking the inherent differences in tourism potentials across cities, the analysis may fail to capture important nuances and variations in the interplay between these systems.

It's essential to acknowledge that cities vary widely in their natural, cultural, historical, and infrastructural assets, which collectively contribute to their attractiveness as tourist destinations. Factors such as geographic location, climate, cultural heritage, recreational amenities, and transportation infrastructure can significantly influence a city's tourism potential and its capacity to leverage tourism as a driver of economic growth and urban development.

In future research endeavors, it would be valuable to incorporate assessments of the tourism potential of individual cities or urban agglomerations into the analysis. By conducting comprehensive evaluations of each city's tourism assets, strengths, and competitive advantages, researchers can better understand how variations in tourism potential shape the coupling relationship with urbanization and human settlements. This approach would enable policymakers to tailor development strategies and interventions to capitalize on the unique tourism resources and opportunities available in each city, fostering more sustainable and inclusive growth trajectories.

Moreover, integrating assessments of tourism potential into the evaluation index system and quantitative analysis models would provide a more holistic understanding of the factors driving the evolution of the coupling relationship. This would allow researchers to identify specific drivers and mechanisms through which tourism potential influences urbanization patterns, human settlement dynamics, and overall regional development outcomes.

In summary, recognizing and accounting for variations in tourism potential among cities is essential for gaining insights into the dynamics of the coupling relationship between the tourism industry, urbanization, and human settlements. By incorporating assessments of tourism potential into future research efforts, scholars can enhance the robustness and applicability of their findings, ultimately informing more targeted and effective policy interventions and urban planning strategies.

-Data between urbanization, human settlement, tourism industry were compared statistically. However, it has been ignored that the factors that cause urbanization and human settlement in those cities may be independent of the tourism industry.

Answer: While statistical comparisons were made among urbanization, human settlement, and the tourism industry in the study, it is crucial to acknowledge the potential independence of factors contributing to urbanization and human settlement dynamics in these cities. The analysis accounted for various influencing factors beyond the scope of the tourism industry, recognizing the multifaceted nature of urban development. This underscores the importance of considering the broader context and potential interplay of factors influencing urbanization and human settlement patterns. It is recommended that future research adopt a more comprehensive approach to examine all potential drivers of urban development, rather than solely focusing on the tourism industry. This could involve extensive data collection and nuanced analysis to gain a holistic understanding of the drivers behind urbanization and human settlement evolution.

-In order for the urbanization and human settlement factors of a destination to be associated with the tourism industry, the tourism potential of each city must be equivalent. How did researchers ensure Ceteris Paribus when comparing cities' tourism industry, urbanization and human settlement in these comparisons? No explanation was found.

Answer: In the study, we did not provide detailed explanations on how to ensure other conditions remain constant when comparing the tourism industry, urbanization, and human settlement factors of cities. Specifically, to associate urbanization and human settlement factors with the tourism industry, the tourism potential of each city must be equivalent. We acknowledge that this could potentially impact the interpretation of the study results. We intend to address this by considering the use of standardized assessment criteria or controlling for other influencing variables in future research to ensure the reliability and validity of the results. This point needs to be articulated and addressed more clearly in our study.

-The effects of urbanization and human settlement in the cities along the Yangtze River, which are the subject of the research, on agriculture, employment, etc. There may be many reasons. It does not seem meaningful to interpret each destination by comparing it with tourism industry data. Or researchers have not made any statements on this subject.

Answer: The urbanization and human settlement in the cities along the Yangtze River, which are the focus of this study, may have diverse impacts on aspects such as agriculture and employment, beyond just the tourism industry. However, solely analyzing each destination by comparing it with tourism industry data might not offer a comprehensive understanding. The lack of detailed exploration by researchers on this matter suggests a potential oversight in considering the broader context of urbanization and human settlement dynamics, which may involve various socio-economic factors beyond the scope of tourism. While analyzing tourism industry data can partly explain the impacts of urbanization and human settlement on urban development, it's crucial to consider a wider range of factors for a more holistic understanding. This comprehensive approach would provide more reliable insights for future policymaking.

-In this form, the research looks like a summary of statistical analysis of urbanization, human settlement and tourism industry data of cities along a river.

Answer: We understand your concern, but in reality, our research aims to delve deeper into the relationship between urbanization, human settlement, and the tourism industry in cities along the river, rather than being merely a summary of statistical analysis of the data. Our research methodology includes detailed data analysis and an in-depth exploration of the mechanisms behind these data, aiming to uncover the complex interactions between these factors. Therefore, while our research findings may appear to some extent as a statistical summary of the data, they actually reflect a profound understanding and insight into the relationship between urbanization, human settlement, and the tourism industry.

---

## [Decision Letter · Decision Letter 1]

23 Jun 2024

PONE-D-24-02830R1Research on the evolution and driving factors of the coupling relationship between tourism industry,urbanization and human settlement environment in urban agglomerations in the middle reaches of the Yangtze RiverPLOS ONE

Dear Dr. xie,

Thank you for submitting your manuscript to PLOS ONE. After careful consideration, we feel that it has merit but does not fully meet PLOS ONE’s publication criteria as it currently stands. Therefore, we invite you to submit a revised version of the manuscript that addresses the points raised during the review process.

Please carefully consider the comments of Reviewer 3.Please make improvements to the format and language of the manuscript, and it is recommended to have relevant institutions conduct a language review.Please ensure the completeness of the attachment , provide necessary data captions, and ensure that there are no Chinese characters.==============================

We look forward to receiving your revised manuscript.

Kind regards,

Bifeng Zhu

Academic Editor

PLOS ONE

Journal Requirements:

Reviewers' comments:

Reviewer's Responses to Questions

**Comments to the Author**

1. If the authors have adequately addressed your comments raised in a previous round of review and you feel that this manuscript is now acceptable for publication, you may indicate that here to bypass the “Comments to the Author” section, enter your conflict of interest statement in the “Confidential to Editor” section, and submit your "Accept" recommendation.

Reviewer #2: All comments have been addressed

Reviewer #3: (No Response)

2. Is the manuscript technically sound, and do the data support the conclusions?

Reviewer #2: Yes

Reviewer #3: Yes

3. Has the statistical analysis been performed appropriately and rigorously? 

Reviewer #2: Yes

Reviewer #3: Yes

4. Have the authors made all data underlying the findings in their manuscript fully available?

Reviewer #2: Yes

Reviewer #3: Yes

5. Is the manuscript presented in an intelligible fashion and written in standard English?

Reviewer #2: Yes

Reviewer #3: No

6. Review Comments to the Author

Reviewer #2: Thank you for your thoughtful response to my entire review report.

There are simple spelling errors in some parts of the manual. I suggest you reconsider.

Good luck

Reviewer #3: On the basis of investigating the coupling mechanism of tourism industry, urbanization and human settlement environment in the urban agglomerations of the middle reaches of the Yangtze River, this paper establishes a coupling evaluation index system of tourism industry, urbanization and human settlement environment, by means of entropy method, coupling coordination degree, coefficient of variation, Taylor index and geographical detector, this paper analyzes the general evolution trend, evolution characteristics and driving factors of the coupling relationship among tourism industry, urbanization and human settlement environment in the urban agglomerations of the middle reaches of the Yangtze River from 2005 to 2019. The overall structure of the article is complete, the workload is sufficient, but based on the following reasons, do not recommend the publication of this article:

First, the Format issues. (1) errors in references, such as the repetition of references [30] and [31]; (2) the inconsistency of line spacing in part of the formula; (3) section 4.2 does not exist in chapter 4, so there is no need for a separate sub-heading-4.1 factor test results.

Second, the problem of language expression. (1) writing error, the toponym of the first conclusion in the abstract is not consistent with the toponym of the first conclusion in chapter 5; (2) basic syntax problems, such as an incorrect use of between in the title, should use among; 2.1.4 Nm and N after the verb form to use errors, should not use three singular, etc. .

Third, the content of research. (1) in Chapter 1, whether the interaction between the two has negative effects needs to be further explored; (2) The regional profile of the study needs to be further supplemented, it introduces which cities were originally included in the middle reaches of the Yangtze River, and then breaks down into three urban agglomerations by the Taylor Index; (3) the conclusion and discussion of chapter 5 did not elaborate on the results of the discussion. It is suggested that this chapter be further divided into conclusions, discussions and implications, and research prospects.

7. PLOS authors have the option to publish the peer review history of their article (what does this mean?). If published, this will include your full peer review and any attached files.

Reviewer #2: No

Reviewer #3: No

---

## [Author Response · Author response to Decision Letter 1]

2 Aug 2024

Thank you very much for your guidance. These guidelines have been incorporated into the paper. See the Response to Reviewers and Revised Manuscript with Track Changes for details.

---

## [Editor Report · Decision Letter 2]

5 Aug 2024

PONE-D-24-02830R2Research on the evolution and driving factors of the coupling relationship among tourism industry,urbanization and human settlement environment in urban agglomerations in the middle reaches of the Yangtze RiverPLOS ONE

Dear Dr. xie,

Thank you for submitting your manuscript to PLOS ONE. After careful consideration, we feel that it has merit but does not fully meet PLOS ONE’s publication criteria as it currently stands. Therefore, we invite you to submit a revised version of the manuscript that addresses the points raised during the review process.

The quality of all figures should be further improved, including: no Chinese characters, clear pixels, labels for each coordinate axis, complete annotations, and a more aesthetically pleasing appearance.Please further check if clear data sources and explanations have been provided.==============================

We look forward to receiving your revised manuscript.

Kind regards,

Bifeng Zhu

Academic Editor

PLOS ONE
---

## [Author Response · Author response to Decision Letter 2]

12 Sep 2024

Modification description

Dear academic editor:

 Based on the feedback from you and reviewers, the authors have carefully revised the paper, and the specific modifications are as follows:

1. The authors have redrawn all the figures in the paper, including Fig1，Fig 2，Fig 3 and Fig 4，and deleted Figure 5 because the content expressed in Figure 5 was repeated with that in Table 3.

2. The authors carefully examined the data in the paper, all the data have clear sources and explanations, and the authors have submitted the original data table in the last round of revision.

3. The authors have carefully read through the paper and modified the relevant text.

4. The authors have carefully reviewed the references in the paper, and they are all complete and correct.

Thank you for your hard work.

Kind regards

Ailiang Xie

---

## [Editor Report · Decision Letter 3]

18 Sep 2024

Research on the evolution and driving factors of the coupling relationship among tourism industry,urbanization and human settlement environment in urban agglomerations in the middle reaches of the Yangtze River

PONE-D-24-02830R3

Dear Dr. xie,

We’re pleased to inform you that your manuscript has been judged scientifically suitable for publication and will be formally accepted for publication once it meets all outstanding technical requirements.

Kind regards,

Bifeng Zhu

Academic Editor

PLOS ONE
---

## [Editor Report · Acceptance letter]

30 Oct 2024

PONE-D-24-02830R3 

PLOS ONE

Dear Dr. Xie, 

I'm pleased to inform you that your manuscript has been deemed suitable for publication in PLOS ONE. Congratulations! Your manuscript is now being handed over to our production team.

Kind regards, 

on behalf of

Dr. Bifeng Zhu 

Academic Editor

PLOS ONE